# Plasma Proteomic Analysis in Non-Small Cell Lung Cancer Patients Treated with PD-1/PD-L1 Blockade

**DOI:** 10.3390/cancers13133116

**Published:** 2021-06-22

**Authors:** Mohamed Eltahir, Johan Isaksson, Johanna Sofia Margareta Mattsson, Klas Kärre, Johan Botling, Martin Lord, Sara M. Mangsbo, Patrick Micke

**Affiliations:** 1Department of Immunology, Genetics and Pathology, Uppsala University, 751 85 Uppsala, Sweden; mohamed.eltahir@igp.uu.se (M.E.); johan.isaksson@regiongavleborg.se (J.I.); johanna.mattsson@igp.uu.se (J.S.M.M.); johan.botling@igp.uu.se (J.B.); 2Department of Pharmaceutical Biosciences, Science for Life Laboratory, Uppsala University, 751 24 Uppsala, Sweden; martin.lord@farmbio.uu.se (M.L.); sara.mangsbo@farmbio.uu.se (S.M.M.); 3Centre for Research and Development, Uppsala University, Region Gävleborg, 801 88 Uppsala, Sweden; 4Department of Microbiology, Cell and Tumor Biology, Karolinska Institute, 171 77 Stockholm, Sweden; Klas.Karre@ki.se

**Keywords:** lung cancer, biomarkers, liquid biopsy, immune checkpoint inhibitors, PD-L1

## Abstract

**Simple Summary:**

Immunotherapy leads to highly variable responses in lung cancer patients. We assessed the value of a blood-based test to predict which patients would benefit from this new treatment modality. We determined that some patients have higher and lower levels of immune markers in their blood samples, and that this is related to better survival without tumor growth. The blood test has the potential to help select the optimal therapy for lung cancer patients.

**Abstract:**

Checkpoint inhibitors have been approved for the treatment of non-small cell lung cancer (NSCLC). However, only a minority of patients demonstrate a durable clinical response. PD-L1 scoring is currently the only biomarker measure routinely used to select patients for immunotherapy, but its predictive accuracy is modest. The aim of our study was to evaluate a proteomic assay for the analysis of patient plasma in the context of immunotherapy. Pretreatment plasma samples from 43 NSCLC patients who received anti-PD-(L)1 therapy were analyzed using a proximity extension assay (PEA) to quantify 92 different immune oncology-related proteins. The plasma protein levels were associated with clinical and histopathological parameters, as well as therapy response and survival. Unsupervised hierarchical cluster analysis revealed two patient groups with distinct protein profiles associated with high and low immune protein levels, designated as “hot” and “cold”. Further supervised cluster analysis based on T-cell activation markers showed that higher levels of T-cell activation markers were associated with longer progression-free survival (PFS) (*p* < 0.01). The analysis of single proteins revealed that high plasma levels of CXCL9 and CXCL10 and low ADA levels were associated with better response and prolonged PFS (*p* < 0.05). Moreover, in an explorative response prediction model, the combination of protein markers (CXCL9, CXCL10, IL-15, CASP8, and ADA) resulted in higher accuracy in predicting response than tumor PD-L1 expression or each protein assayed individually. Our findings demonstrate a proof of concept for the use of multiplex plasma protein levels as a tool for anti-PD-(L)1 response prediction in NSCLC. Additionally, we identified protein signatures that could predict the response to anti-PD-(L)1 therapy.

## 1. Introduction

The discovery of immune checkpoint inhibitors (ICIs) that target the programmed death-1/programmed death-ligand-1 (PD-1/PD-L1) axis has revolutionized the treatment of non-small cell lung cancer (NSCLC). Currently, several ICIs are approved for use in monotherapy or in combination with chemotherapy as first- or second-line treatments [1,2,3,4,5]. However, primary or secondary resistance is common, and only around 20% of advanced NSCLC patients develop durable clinical responses from anti-PD-(L)1 therapy [2,3,4,6]. Furthermore, adverse events, which are sometimes life-threatening, can occur and prevent therapy continuation, even in responsive patients [7]. Therefore, accurate prediction of the response to therapy is of high clinical relevance.

Today, immunohistochemical analysis of PD-L1 expression in tumor cells is routinely used as a companion or complementary diagnostic before patient treatment. Assays are performed with tissue biopsies of advanced NSCLC patients, providing a PD-L1 tumor proportion score that guides the clinical use of ICIs [8]. Indeed, high PD-L1 expression in the tumor correlates with a higher response and prolonged patient survival. Nevertheless, many patients with high PD-L1 expression do not achieve desirable antitumor responses, and patients who lack PD-L1 expression may still show a clinical response [9]. The performance of the biomarker is aggravated by pre-analytical and technical issues and the subjective nature of the microscopic interpretation of PD-L1 staining [10,11,12]. Furthermore, PD-L1 expression tissue heterogeneity contributes to the uncertainty of the assay results. As an alternative, the tumor mutation burden (TMB), the number of mutations present in the tumor per coding area, has been suggested as a better biomarker for predicting anti-PD-(L)1 therapy outcomes across many cancers, including NSCLC [13,14]. Recently, pembrolizumab was approved as an agnostic drug for the treatment of solid tumors with ≥10 mutations/megabase [15]. However, the predictive performance of TMB is similar to that of PD-L1 expression, and the cost, standardization, and complexity of next-generation sequencing analysis limit its use in diagnostic practice [16].

In parallel with tumor tissue-based biomarkers, blood-based biomarkers have been evaluated for their ability to provide systemic information about individual responsivity to anti-PD-(L)1 therapy. The non-invasive nature and general accessibility of blood collection present further advantages that make repeated sampling possible [17]. For example, conventional immune cell counts, such as a high absolute neutrophil-to-lymphocyte count at baseline, have been associated with shorter progression-free survival (PFS) and overall survival (OS) with anti-PD-1 therapy in NSCLC [18]. Furthermore, low baseline soluble PD-L1 levels have been associated with a superior objective response rate with nivolumab in NSCLC, melanoma, and renal cell cancer [19,20,21], and lower baseline interleukin 6 (IL-6) levels have been associated with longer PFS and OS in NSCLC patients treated with anti-PD-(L)1 therapy [22]. In another study, responders to nivolumab had high plasma granzyme B levels when nivolumab treatment was started [23]. Despite promising indications in some studies, most have been restricted to only one or a few markers. A holistic strategy designed to identify cancer immunity patterns might increase the detection of potential associations and demonstrate higher predictive accuracy. 

In this study, we used a multiplex plasma proteomic platform to analyze the expression profile of 92 proteins in blood samples collected from NSCLC patients who received anti-PD-(L)1 therapy. The aim was to assess the diagnostic feasibility of this multiplex proteomic approach and to use it as a proof of concept for the prediction of NSCLC patients’ responses to anti-PD-(L)1 therapy.

## 2. Materials and Methods

### 2.1. Patient Cohorts and Samples

Blood samples were obtained from a patient cohort diagnosed with NSCLC and treated with anti-PD-(L)1 therapy. Plasma samples from 43 NSCLC patients were obtained from the Gävle Biobank. The samples were collected between 2016 and 2019. All patients had samples collected before the initiation of NSCLC systemic therapy (Figure 1A). Tumors were staged based on the AJCC staging system (8th edition). Performance status (PS) was assigned according to the WHO Performance Status score [24]. Patients were grouped according to their best response to treatment. Evaluation of responses was performed in a multidisciplinary clinical setting and verified independently during data collection. Patients with partial response (PR) and stable disease (SD) were considered the clinical benefit group and were distinct from patients with progressive disease (PD). There were no patients with a complete response to therapy in the cohort. PFS was defined as the time from the anti-PD-(L)1 therapy start date to the date of confirmed progression, and OS was defined as the time from the immunotherapy start date until the date of death. Table 1 summarizes the clinical characteristics of the patients included in the study.

### 2.2. Sample Analysis

A proximity extension assay (PEA) was performed at the Clinical Biomarker facility at the Science for Life Laboratory (SciLifeLab) in Uppsala, Sweden, using the 92-protein OLINK immune-oncology protein panel (www.OLINK.com, accessed on 15 January 2021). The assay used one microliter of sample for simultaneous multiplex measurement of 92 proteins, of which 84 were above the lower detection limit of the assay (Appendix A). Data were exported as Normalized Protein eXpression (NPX) values, a relative unit on a log2 scale. An NPX value does not represent an absolute protein level but rather relative expression; a high NPX value indicates a high protein level and vice versa.

### 2.3. Statistical Analysis

Student’s *t*-test with correction for multiple testing using the Benjamini-Hochberg procedure was used to interrogate differences in NPX levels between two groups. Statistical summary of relevant clinical parameters was performed using Fisher’s exact, Mann-Whitney, or chi-square test. The effect of candidate proteins on survival was investigated using multivariate Cox regression analysis with the log-rank test to compare the survival curves of two or more groups in Kaplan-Meier plots. The discriminative power of the selected proteins for binary outcomes was evaluated by univariate and multivariate logistic regression models. A random forest (RF) classifier model based on all 84 proteins was constructed for comparison; 10-fold cross-validation was repeated 10 times with an optimal model obtained at mtry of 2 for 1000 trees. The RF model was trained on 70% of the data and validated on the remaining 30%. All other models were based on all the samples (*n* = 43). Statistical analyses were performed using R v4.0.3 (R Core Team (2020); R: A language and environment for statistical computing; R Foundation for Statistical Computing, Vienna, Austria; https://www.R-project.org/, accessed on 15 January 2021). The R packages “survival”, “survminer”, “caret” and “pROC” were used for the survival analyses and binary classification models.

### 2.4. Ethical Considerations

Sampling and sample analysis were approved by the Swedish Ethical Review Authority (DNR 2010/198 and 2017/076).

## 3. Results

### 3.1. Patient Cohorts Clinical and Sample Characteristics

In total, 43 plasma samples were obtained from a cohort of 43 advanced NSCLC patients treated with anti-PD-(L)1 therapy. Samples were obtained at a median of 15 days after NSCLC diagnosis and a median of 205.5 days before the start of anti-PD-(L)1 therapy (Figure 1A,B). The patients’ clinical characteristics are summarized in Table 1. Of all treated patients, 18 (41.9%) had PD, 6 (14.0%) had SD, and 11 (25.6%) had PR. It was not possible to assess the clinical response in eight patients (18.6%) due to either interruption of therapy or loss of follow-up. The median PFS and OS were 4.7 months and 10.7 months, respectively.

### 3.2. Proteomic Signatures Detected in the Plasma of NSCLC Patients

To characterize the soluble peripheral blood protein profiles in NSCLC patients before anti-PD-(L)1 therapy, we performed a multiplex PEA-based analysis of 92 plasma proteins (Appendix A). Of the 92 proteins, levels of eight proteins were below the assay detection limit and were therefore excluded from further analysis. Unsupervised hierarchical clustering of the remaining 84 proteins revealed two patient clusters with distinct protein profiles. The first cluster (*n* = 25) patients generally had higher immune and cancer-related protein levels, while the second cluster (*n* = 18) patients demonstrated lower protein levels. With an analogy to local cancer immune response patterns, we defined the protein signatures as immune “hot” and immune “cold” (Figure 2A). The clusters were not associated with differences in tumor type, stage, sex or age, PD-L1 expression in the tumor, response type, or survival. To further investigate this pattern, a focused supervised hierarchical clustering model with the data portioned into two groups by k-means clustering was used with selected protein subgroups. Based on the immune pathway involved and cellular expression, we defined the following protein panels: T-cell activation-related (22 proteins), antigen-presenting cell-related (APC-related; 9 proteins), myeloid/angiogenesis-related (25 proteins), and tumor-related (9 proteins; Appendix A).

The focused analysis confirmed the observation of the existence of “hot” and “cold” archetypes. This was particularly pronounced for the T-cell activation markers (Figure 2B, Table 2 and Appendix A).

Notably, patients with “hot” profiles demonstrated significantly longer PFS times than patients with “cold” profiles (*p* = 0.006; Figure 3A). The significance was maintained in the multivariate analysis (HR = 0.29, CI = 0.11–0.79, *p* = 0.015) adjusted to the covariates of tumor stage, gender, age, and PS (Figure 3B). These findings illustrate the clinical impact and independence of the plasma profiles, although this benefit in PFS did not translate into differences in OS (*p* = 0.58; Appendix A). To further explore specific protein classes, we split the cohort into two groups, using supervised clustering, based on APC- and myeloid/angiogenesis-related proteins and tumor-related proteins (Appendix A). Notably, also for APC-related proteins (*p* = 0.13) and myeloid/angiogenesis-related proteins (*p* = 0.05), a clear trend toward better survival was observed. However, these associations were not observed in the analysis of OS (Appendix A). When the tumor-related proteins were analyzed, the “hot” signature tended to be associated with shorter PFS (*p* = 0.13) and shorter OS (*p* = 0.08; Appendix A).

### 3.3. Markers of Response to Anti-PD-(L)1 Therapy and a Response-Predictive Signature

Of all available clinical (including age, gender, stage, and PS) and histopathological data, only the expression of PD-L1 in tumor cells (tumor proportion score) detected using immunohistochemical means was associated with the response to immunotherapy. The clinical benefit group (PR and SD) demonstrated higher PD-L1 expression in tumor cells (mean = 57.7%, SD = 30.5) than the non-responder PD group (mean = 13.2%, SD = 25.4; *p* < 0.001, false discovery rate (FDR) = 0.023; Figure 4A). In the clinical benefit group, 12 out of 16 cases were positive for PD-L1, whereas 5 out of 18 cases in the PD group had PD-L1 ≥ 1% (*p* = 0.002, chi-square test). This difference was mainly based on patients with PR and PD (Appendix A).

In an explorative analysis, we compared the pretreatment plasma protein levels in the clinical benefit and PD groups to identify differentially expressed proteins. We identified five proteins with differential plasma levels (Appendix A). Three proteins associated with T-cell activation, C-X-C motif chemokine ligand 10 (CXCL10), CXCL9, and interleukin-15 (IL-15), were present at higher levels in the clinical benefit group compared to the PD group (*p* < 0.05; Figure 4B–D). On the contrary, the pretreatment plasma levels of two proteins, caspase-8 (CASP8) and adenosine deaminase (ADA), were higher in the progression group (Figure 4E,F). However, after rigorous correction for multiple testing using the Benjamini-Hochberg procedure, none of the proteins described reached an FDR < 0.05. This was most likely due to the small sample size. Thus, we consider these results to be descriptive.

Logistic regression-based classification models for proteins that were significantly different between the clinical benefit and PD groups (i.e., CXCL9, CXCL10, IL-15, CASP8, and ADA) were constructed to investigate the potential of the proteins as discriminative predictive biomarkers for patients responding to PD-(L)1 inhibitors. An RF classification model with all 84 proteins based on 1000 trees and a mtry of 2 was additionally constructed for comparison. Linear models were built for each protein separately and in combination and for tumor PD-L1 levels. For the respective models, receiver operating characteristic (ROC) curves were constructed (Figure 4G). The area under the curve (AUC) value for each of the five proteins ranged from 68–77%, with the highest value for IL-15 at 76.7%. The same AUC value was obtained for tumor PD-L1 expression: 75.5%. However, the five-protein combination improved the model’s ability to discriminate between the clinical benefit and PD groups, with an AUC of 94.1% (Figure 4G). In this instance, we also observed that the model based on the combination of the five proteins (CXCL9, CXCL10, IL-15, CASP8, and ADA) performed better than the RF classifier at 83.3% (Figure 4G). It should be noted that caution must be taken with overfitting regarding the predictive power of the models due to the relatively small cohort size and that the linear models were built on the whole cohort compared to the RF model. Interestingly though, out of the 84 proteins computed to construct the RF model, CXCL10 was identified as the protein with the highest discriminating power when ranked by variable importance (VIMP) in the model. Additionally, CXCL9, IL-15, CASP8, and ADA were also among the 10 most important classification variables in the model (Appendix A).

### 3.4. Plasma Proteins as Survival Predictors

Next, we studied the association between plasma immuno-oncology protein levels and survival. As a reference, patients with high PD-L1 expression in tumors determined by immunohistochemistry (IHC) demonstrated longer PFS (*p* = 0.01; Appendix A).

Similarly, when examining individual plasma protein levels with age, stage, gender, and PS as covariates in Cox regression analysis, we observed an association between plasma MUC16, CASP8, CXCL10, and TNFRSF14 levels and PFS. However, when these significant markers were combined with PD-L1 IHC expression in a single multivariate analysis with the same covariates, the only tumor-related protein found to have a significant association with the risk of progression in patients was MUC16; it was found at levels significantly above the median (HR = 2.01, CI = 1.26–3.20, *p* = 0.004; Figure 5A). In this multivariate analysis, PD-L1 expression in tumors (1–50% and 50–100%) was significantly associated with a lower risk of tumor progression over 12 months; this was not observed for patients with tumors that scored in the range of 0–1% (Figure 5A).

Regarding OS, when examining the relationship between the individual plasma protein levels and OS with age, stage, gender, and PS as covariates in a Cox regression analysis, associations between OS and CD224, IL-6, TRAIL, MUC16, ADA, IL-18, GZMH, TNFSF14, FASLG, DCN, ANGPT2, IFNG, CASP8, and soluble plasma PD-L1 were observed. However, combining these significant markers along with tumor PD-L1 expression in a single multivariate Cox regression with the covariates of stage, age, gender, and PS revealed a correlation between longer OS and high plasma FASLG levels, and between shorter OS and high plasma CD244, MUC16, IL18, and ANGPT2 levels (Figure 5B). Again, high tumor PD-L1 expression (50–100%) was associated with favorable OS (HR = 3.8 × 10^−4^, CI = 5.2 × 10^−6^–2.8 × 10^−2^, *p* < 0.001).

## 4. Discussion

This study presents a plasma proteomic approach for the characterization of proteins in the context of ICI treatment. As a proof of concept, this technique allows the determination of the presence of a multitude of proteins, either as single markers or as composite signatures that predict response to immunotherapy and patient survival. Our results indicate that systemic immune markers demonstrate distinct patterns of clinical relevance; thus, they can be considered as an alternative for patient stratification. The identified proteins might also provide mechanistic information to optimize immunotherapeutic interventions.

Due to their capacity to provide readily accessible information, blood-based assays are a promising option to complement or even replace cancer diagnostics that rely on tissue analysis, which has several limitations [25]. Rebiopsies of lung tumors are difficult to perform in the clinical setting, and immunoscoring (e.g., PD-L1 analysis) is mostly done only once at diagnosis; therefore, the results are not applicable for longitudinal monitoring of responses (26). For lung cancer diagnostics, “liquid biopsies” are performed when tissue biopsy procurement is too hazardous or when therapy resistance develops [25,26,27]. Such blood-based analysis aims to identify genomic features with the rationale that tumor DNA is leaking from the cancer tissue into the bloodstream [28]. The identification of cancer-specific mutations in liquid biopsies proves this hypothesis [26,29].

In our proteomic approach, the detected plasma protein levels might also be influenced by protein release from cancer tissue. This leakage can be utilized for detecting tumor-specific proteins, best exemplified by tumor markers, such as PSA or CEA, for screening purposes or monitoring treatment response [30,31]. On the other hand, particularly when immune markers and cytokines are quantified, plasma protein levels can be a result of the immune response in tumor tissue and in distant lymphoid organs. Therefore, protein profiles most likely reflect a combination of local and systemic cancer immunity. With this background, identifying regulated proteins might provide insight into how the immune system responds to cancer locally and systemically and why immune activation with checkpoint inhibitors is effective in some patients and not others.

The most intriguing finding is probably the identification of two distinct systemic patterns of immune-related markers in this cohort. In the unsupervised cluster analysis, we identified “hot” and “cold” immune activation patterns. This clustering encouraged further analysis of T-cell activation-related markers that were independently associated with PFS. The concepts of immune “hot” and immune “cold” are derived from the immune cell infiltration patterns in cancer tissue, often defined by CD8^+^ T cell-rich tumors and the upregulation of T-cell activation markers [32]. These tumors have a generally favorable prognosis and respond better to immunotherapy [33,34]. It is possible that the immune profiles reported here reflect the tumor status and could indicate patient immune status. Although we could only demonstrate a trend toward a better immune response in patients expressing a “hot” pattern, these patients showed significantly improved PFS with immunotherapy when the analysis was restricted to T-cell-related proteins. This finding holds promise that immune profiles can be used to stratify patients for immunotherapy, although independent confirmation is necessary. For this, a comparison of tumor tissue and plasma immune profiles would be critical for the interpretation of our results.

In addition to general protein patterns, we identified five proteins (CXCL9, CXCL10, IL-15, CASP8, and ADA) that, either as single markers or as composite signatures, were associated with the response to checkpoint inhibition. CXCL9 and CXCL10 are two important chemokines belonging to the CXC family induced by INFG [35,36]. They elicit their functions by interacting with the chemokine receptor CXCR3. Both chemokines can be expressed by several cell types, including leucocytes, fibroblasts, and tumor cells. CXCL9 mediates local lymphocyte infiltration into tumor tissue, which suppresses tumor growth [37]. Experimentally, CXCL9-deficient cancer cells have been shown to be more tumorigenic than cancer cells transduced with CXCL9 [38]. Also, CXCL10 demonstrated tumor-inhibiting properties in vivo [39]. Recently, the important role of macrophage-derived CXCL10 and CXCL9 in T-cell infiltration after PD-L1 blockade was determined [40]. Melanoma patients treated with immunotherapy had significantly better survival rates when the tumor expressed CXCL9 or CXCL10 [41]. Our results support the notion that CXCL9 and CXCL10 are also critical in the context of lung cancer immunotherapy.

The identification of IL-15 can be explained by its potent ability to activate T cells and NK cells. Interestingly, an IL-15 superagonist (ALT-803) increases the efficacy of the PD-L1 blockade in murine solid cancer models [42]. This agonist has already been tested in a phase Ib trial with NSCLC patients in combination with nivolumab and showed promising response rates [43].

The levels of CASP8 and ADA were negatively associated with the immune response to checkpoint inhibition. CASP8 mediates apoptosis, and increased expression is observed in many cancer types [44]. ADA is an enzyme involved in purine metabolism and is elevated in plasma from patients with inflammatory diseases and cancer [45,46]. Both enzymes are also involved in immune cell metabolism and may mediate immune regulatory functions [47,48]. More focused studies are required to elucidate if and how high levels of CASP8 and ADA are connected to the immune therapy response. Not surprisingly, we found that the combination of all five markers further increased the predictive performance and outperformed the hitherto best-established marker: the PD-L1 proportion score. We also identified several proteins that were associated with PFS or/and OS of patients treated with immunotherapy. These results are difficult to interpret because the markers can be either of general prognostic relevance or can be predictive for longer survival after treatment. This aspect can only be addressed when plasma samples of a control group with patients not treated with checkpoint inhibitors are analyzed.

To our knowledge, this is the first time that a proteomic approach has been applied to plasma samples of NSCLC patients to evaluate their response to checkpoint inhibitors. Previous studies evaluated single markers or marker panels [19,23,49,50], limiting the analytical power of blood-based diagnostic approaches. Although the results reported herein support the usefulness of the PEA as a tool to stratify and monitor patients receiving immunotherapy, the findings should be regarded as descriptive. The patient cohort was relatively small, with only 43 patients. It should be noted that this real-world cohort may still be a representative immunotherapy cohort, because tumor PD-L1 expression revealed a clear predictive and prognostic impact. However, it is necessary to conduct a study with a completely independent patient cohort to confirm the associations between the identified proteins reported herein. Another point of concern is the retrospective nature of the study. Diagnostic assay evaluation should ideally be conducted in a prospective manner. Future studies should also include a control group of untreated patients to determine whether any of the immune markers are predictive of response to immunotherapy or only prognostic, indicating patient status.

## 5. Conclusions

In conclusion, the PEA-based approach has value for the quantification of multiple immune-related plasma proteins in the context of immunotherapy in NSCLC. It holds promise as a method for predicting the benefit of immunotherapy and for longitudinal monitoring of therapy response. Focused analysis of the markers identified in this study and further optimization of protein panels will extend the potential of this approach as a non-invasive method for guiding treatment in the era of immunotherapy.

## Figures and Tables

**Figure 1 cancers-13-03116-f001:**
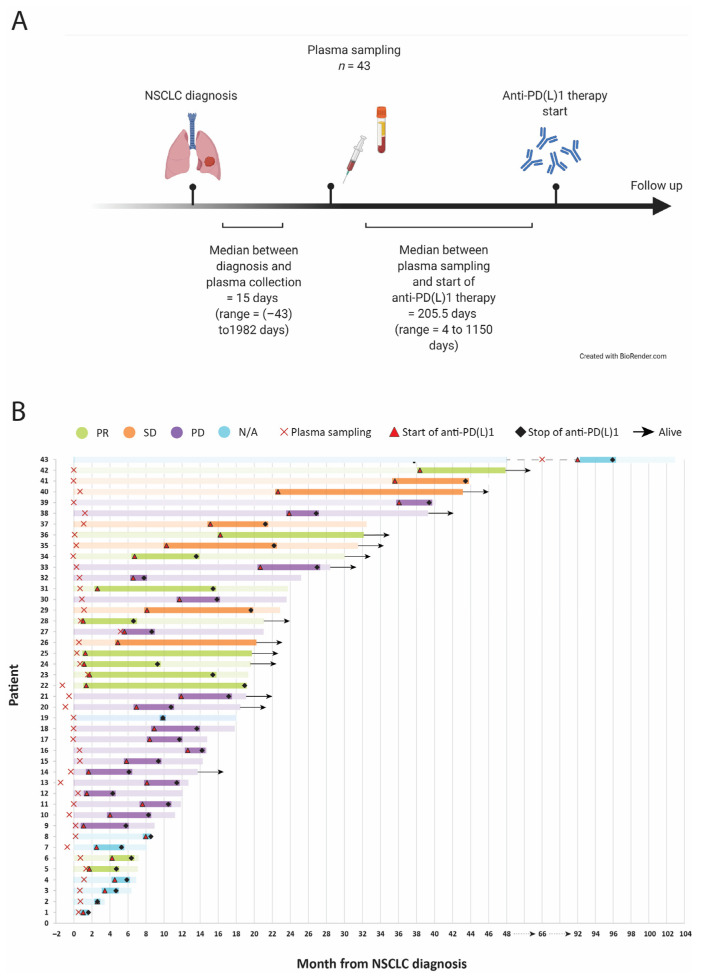
Workflow and clinical events. (**A**) Sequential workflow for diagnosis, sampling, and therapy initiation. The number of samples (*n*), the median days, and the time range between sampling and diagnosis or therapy start are noted. (**B**) Swimmer plot illustrating the individual patients’ clinical courses of treatment start and stop, sampling, and outcome.

**Figure 2 cancers-13-03116-f002:**
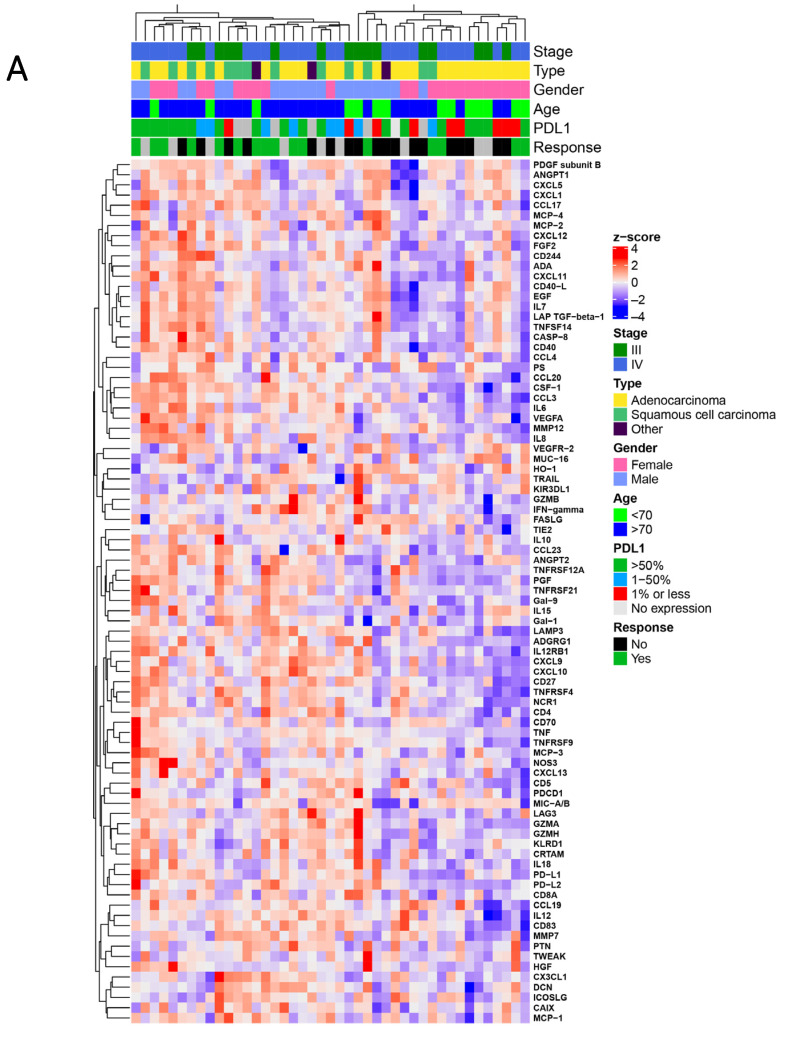
Hierarchical clustering analysis of protein expression. (**A**) Plasma from the patient cohort (*n* = 43, 84 proteins passed quality control). Unsupervised clustering analysis was performed on pre-anti-PD-(L)1 therapy samples based on all the analyzed proteins that were within the assay detection limit (84 proteins). (**B**) K-means clustering analysis for selected T-cell activation-related markers (22 proteins) in the plasma. Normalized protein expression (NPX) is presented as a z-score.

**Figure 3 cancers-13-03116-f003:**
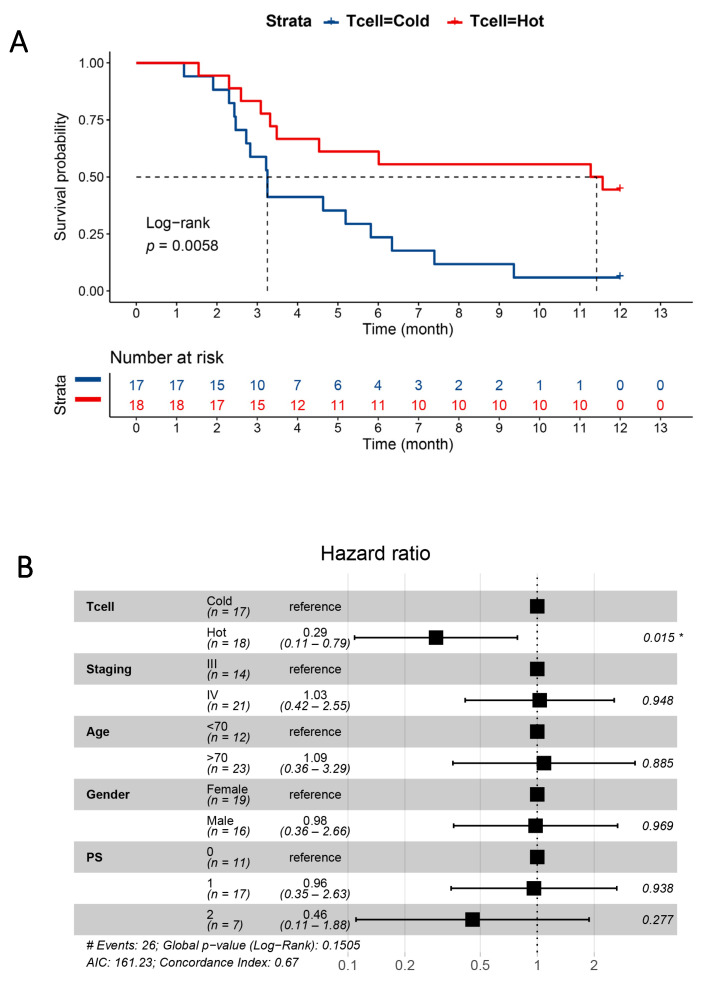
The association between progression-free survival (PFS) and the pretreatment plasma protein signature. (**A**) Kaplan–Meier PFS curves based on hot and cold T-cell activation signatures. A log rank test was performed to compare the two groups. (**B**) Multivariate Cox regression analysis of PFS in the key subgroups. The hazard ratio, 95% confidence interval, and *p*-value for each analysis are displayed in the plot. (* *p* ≤ 0.05, PS—performance status).

**Figure 4 cancers-13-03116-f004:**
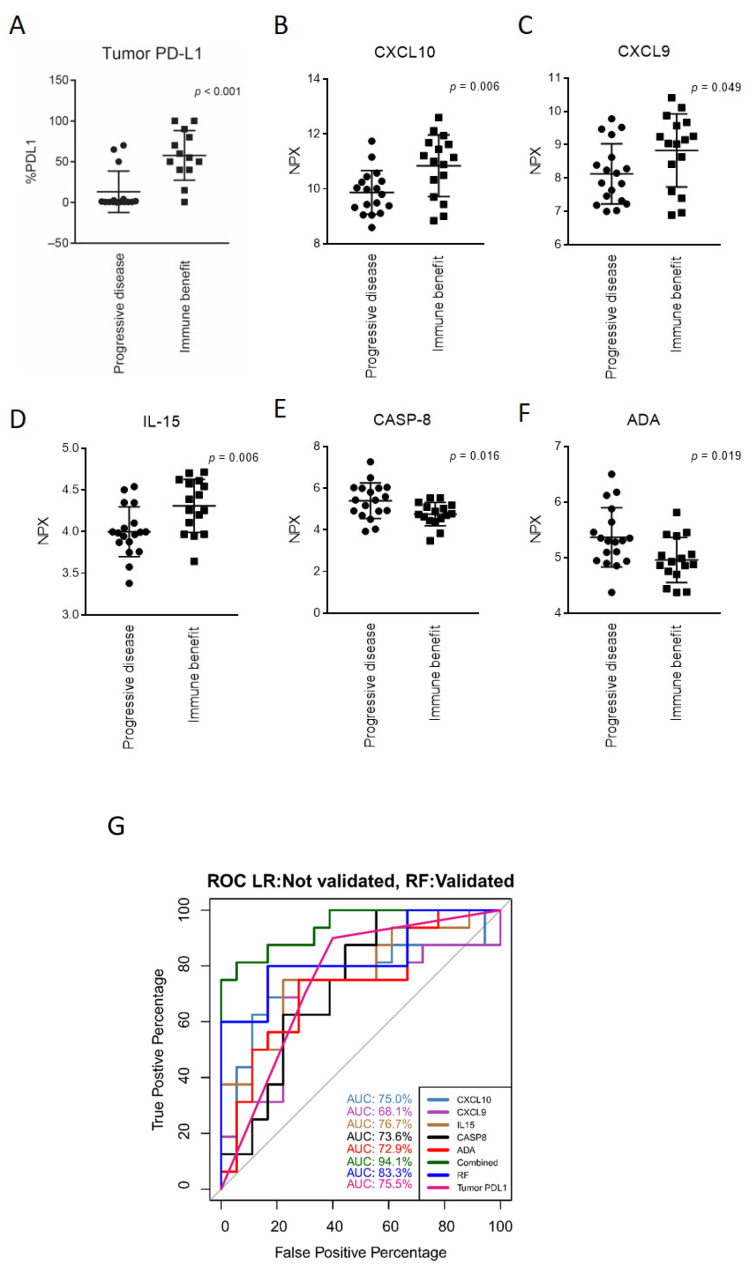
Tumor PD-L1 expression and plasma protein levels at the baseline and the development of the discriminatory model. The population (*n* = 43) was stratified into two groups: immune benefit (including stable disease and partial response) and progressive disease (including patients with progressive disease). (**A**) Comparison of tumor PD-L1 expression, assessed by immunohistochemistry, before the initiation of anti-PD-(L)1 therapy between the immune benefit and progressive disease groups. (**B**–**F**) Plasma proteins with differential expression before the initiation of anti-PD-(L)1 therapy. A two-tailed unpaired t-test or two-tailed non-parametric Mann-Whitney test was used. Only proteins with a *p*-value < 0.05 are displayed. Data are plotted as %PD-L1 or normalized protein expression (NPX). (**G**) Receiver operating characteristic (ROC) curves illustrating the discriminatory model for CXCL10, CXCL9, IL-15, CASP8, and ADA individually or combined and tumor PD-L1. A random forest model containing all 84 proteins was also evaluated.

**Figure 5 cancers-13-03116-f005:**
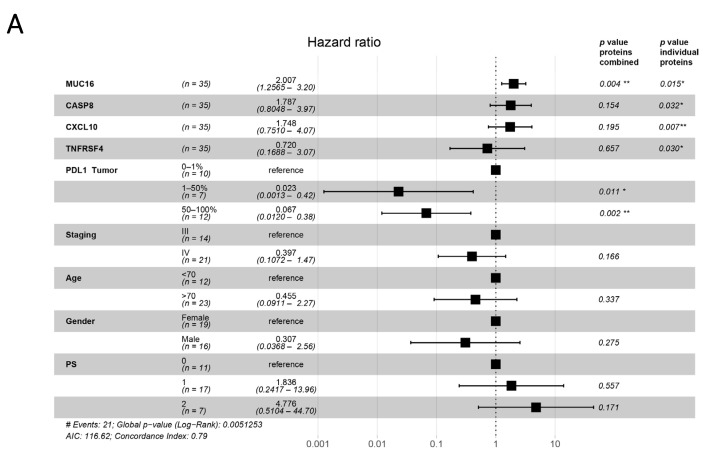
Hazard ratio forest plots. Data are based on multivariate Cox regression analysis for (**A**) progression-free survival (PFS) and (**B**) overall survival (OS). *p*-values are displayed for the multivariate analysis when each protein was analyzed individually and when all the proteins were combined. (* *p* ≤ 0.05, ** *p* ≤ 0.01, *** *p* ≤ 0.001, PS—performance status).

**Table 1 cancers-13-03116-t001:** Demographic and clinical characteristics of the advanced NSCLC patient cohorts included in the study.

Clinical Characteristic	*n* (%)
All patients	43 (100)
Sex	
Female	23 (53.5)
Male	20 (46.5)
Median age (years; at the start of immunotherapy)	72.5 ± 8.8
Smoking status (at diagnosis)	
Ever smoker (smoker/ex-smoker)	39 (90.7)
Never smoker	4 (9.3)
Tumor type	
Adenocarcinoma	27 (62.8)
Squamous cell carcinoma	13 (30.2)
Other	3 (7.0)
Stage	
III	16 (37.2)
IV	27 (62.8)
Performance status	
0 1 2	11 (25.6) 22 (51.2) 10 (23.2)
PD-L1	
No expression	1 (2.3)
<1%	9 (20.9)
1–49% 50–100%	8 (18.6) 19 (44.2)
N/A	6 (14.0)
Therapeutic agent	
Atezolizumab	7 (16.3)
Durvalumab	1 (2.3)
Nivolumab	12 (27.9)
Pembrolizumab	23 (53.5)
Therapy line at which immunotherapy was initiated	
1st line	13 (30.2)
2nd line	19 (44.2)
3rd line	9 (20.9)
4th line	1 (2.3)
5th line	1 (2.3)
Response	
Progressive disease (PD)	18 (41.9)
Stable disease (SD)	6 (14.0)
Partial response (PR)	11 (25.6)
Complete remission (CR)	0 (0.0)
N/A	8 (18.5)
Clinical benefit (SD, PR) Progressive disease (PD) N/A	17 (39.6) 18 (41.9) 8 (18.5)
Survival (from the start of immunotherapy)	
Median overall survival (days) ± SD	315 ± 198.1
Median progression-free survival (days) ± SD	141 ± 202.5

N/A, not applicable.

**Table 2 cancers-13-03116-t002:** Comparison of patients’ clinical characteristics between the T-cell activation-related plasma protein (22 proteins) signature (immune “hot” vs. immune “cold”) subgroups. *p*-values were obtained using Fisher’s exact, Mann-Whitney, or chi-square test.

Clinical Characteristic	Immune “Hot”	Immune “Cold”	*p*-Value
	Protein Signature	Protein Signature	
	*n* = 24 (55.8%)	*n* = 19 (44.2%)	
Sex			0.03
Female	9 (37.5)	14 (73.7)	
Male	15 (62.5)	5 (26.3)	
Median age (years; at the start of immunotherapy)	74.5	68.0	0.005
Smoking status (at diagnosis)			>0.999
Ever smoker (smoker/ex-smoker)	22 (91.7)	17 (89.5)	
Never smoker	2 (8.3)	2 (10.5)	
Tumor type			0.674
Adenocarcinoma	15 (62.5)	12 (63.2)	
Squamous cell carcinoma	8 (33.3)	5 (26.3)	
Other	1 (4.2)	2 (10.5)	
Stage			0.752
III	6 (31.6)	10 (41.7)	
IV	13 (68.4)	14 (58.3)	
Performance status			0.203
0 1 2	4 (16.7) 15 (62.5) 5 (20.8)	7 (36.8) 7 (36.8) 5 (26.3)	
PD-L1			0.197
No expression	0 (0.0)	0 (0.0)	
<1%	3 (12.5)	7 (36.8)	
1–49% 50–100%	6 (25.0) 11 (45.8)	4 (21.1) 6 (31.6)	
N/A	4 (15.8)	2 (10.5)	
Therapeutic agent			0.643
Atezolizumab	3 (12.5)	4 (21.1)	
Durvalumab	1 (4.2)	0 (0.0)	
Nivolumab	6 (25.0)	6 (31.6)	
Pembrolizumab	14 (58.3)	9 (47.4)	
Therapy line at which immunotherapy was initiated			0.246
1st line	10 (41.7)	3 (15.8)	
2nd line	9 (37.5)	10 (52.6)	
3rd line	4 (16.7)	5 (26.3)	
4th line	1 (4.2)	0 (0.0)	
5th line	0 (0.0)	1 (5.3)	
Response			0.065
Progressive disease (PD)	6 (25.0)	12 (63.2)	
Stable disease (SD)	5 (20.8)	1 (5.3)	
Partial response (PR)	7 (29.2)	4 (21.1)	
Complete remission (CR)	0 (0.0)	0 (0.0)	
N/A	6 (33.3)	2 (10.5)	
Clinical benefit (SD, PR, CR) progressive disease (PD) N/A	12 (50.0) 6 (25.0) 6 (25.0)	4 (21.1) 12 (63.2) 3 (15.8)	0.020
Survival (from the start of immunotherapy)			
Median overall survival (days) ± SD	402.5 ± 213.0	236.0 ± 178.8	-
Median progression-free survival (days) ± SD	106.0 ± 125.9	98.5 ± 56.0	-

N/A, not applicable.

## Data Availability

The raw data are available upon reasonable request.

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
