# Peer review of "Plasma Proteomic Analysis in Non-Small Cell Lung Cancer Patients Treated with PD-1/PD-L1 Blockade"

_cancers, 2021, doi:10.3390/cancers13133116_

Round 1

Reviewer 1 Report

In this study, Eltahir M and colleagues used a multiplex plasma proteomic platform to analyze the expression profile of 92 proteins in blood samples collected from NSCLC patients who received anti-PD-(L)1 therapy. The aim of this paper was to assess the diagnostic feasibility of this multiplex proteomic approach and as a proof-of-concept if the assay can predict response to anti-PD-(L)1 therapy in NSCLC. The focused analysis of markers and further optimization of protein panels will further extend the potential as a non-invasive assay to guide treatment in the ear of immunotherapy.

To my eyes, the paper is excellent looking at the proteomic approach applied and the samples of NSCLC patients were used to evaluate the response to checkpoint inhibitors. Moreover, this article helps to identify the Proximity extension assay (PEA)-based assay as a valuable tool to quantify multiple immune-related plasma proteins in the context of immunotherapy in NSCLC. Detailed comments of a single study surely fit into the standard quality of our journal and confirm the importance of data analysis and results discussed. The scientific content is good and the English style and language used in the manuscript are fine. Moreover, I appreciate the scientific efforts to organize this paper and I think that the concept of plasma proteomic analysis is well-argued. The resolution of the table of each section and the reference list meets the quality requirements of our journal. Moreover, the results were clearly discussed and corroborated with what is shown. Finally, the data analyses were interpreted in a comprehensible manner. I didn’t observe any remarkable incongruences throughout the text.

I have only two questions before considering the paper accepted:

  • Our patient cohort is relatively small with only 43 patients. How do you justify the fact that this real-world effect could potentially be important and not be statistically significant in terms of predictive and prognostic values?
  • The lack of both independent cohort and control group with untreated patients would be necessary to discriminate the proteomic expression and immune profile markers. Why didn’t you consider?

Author Response

Reviewer 1: In this study, Eltahir M and colleagues used a multiplex plasma proteomic platform to analyze the expression profile of 92 proteins in blood samples collected from NSCLC patients who received anti-PD-(L)1 therapy. The aim of this paper was to assess the diagnostic feasibility of this multiplex proteomic approach and as a proof-of-concept if the assay can predict response to anti-PD-(L)1 therapy in NSCLC. The focused analysis of markers and further optimization of protein panels will further extend the potential as a non-invasive assay to guide treatment in the era of immunotherapy.

To my eyes, the paper is excellent looking at the proteomic approach applied and the samples of NSCLC patients were used to evaluate the response to checkpoint inhibitors. Moreover, this article helps to identify the Proximity extension assay (PEA)-based assay as a valuable tool to quantify multiple immune-related plasma proteins in the context of immunotherapy in NSCLC. Detailed comments of a single study surely fit into the standard quality of our journal and confirm the importance of data analysis and results discussed. The scientific content is good and the English style and language used in the manuscript are fine. Moreover, I appreciate the scientific efforts to organize this paper and I think that the concept of plasma proteomic analysis is well-argued. The resolution of the table of each section and the reference list meets the quality requirements of our journal. Moreover, the results were clearly discussed and corroborated with what is shown. Finally, the data analyses were interpreted in a comprehensible manner. I didn’t observe any remarkable incongruences throughout the text.

Response: We appreciate the positive evaluation.

Reviewer 1: I have only two questions before considering the paper accepted:

* Our patient cohort is relatively small with only 43 patients. How do you justify the fact that this real-world effect could potentially be important and not be statistically significant in terms of predictive and prognostic values?

* The lack of both independent cohort and control group with untreated patients would be necessary to discriminate the proteomic expression and immune profile markers. Why didn’t you consider?

Response: We agree with the reviewer, and the two questions are connected. The relatively small sample size and the lack of a control group limit the interpretation of the results. It is not clear whether the signatures or single proteins that are associated with prolonged progression-free survival or overall survival are predictive for the effect of immunotherapy or are prognostic for an immune profile associated with a generally better clinical outcome. A randomized trial is the optimal approach to address this question. Also, stage IV patients not treated with immunotherapy could serve as a control group. In this case, the patients should be carefully matched to avoid bias based on patient selection.

Unfortunately, we did not have such balanced control samples for this study. We are now in the process of collecting blood samples from patients who have not been treated with immunotherapy and who could thus serve as controls in a follow-up validation study.

It should be noted that we also found single markers or marker signatures for the radiological response as clearly predictive markers. These findings better support the assumption that the blood-based assay can help during patient selection.

However, we consider our data descriptive, and we are in the process of collecting more samples to validate the findings.

Reviewer 2 Report

The manuscript titled "Plasma proteomic analysis in non-small cell lung cancer patients treated with PD1/PD-L1 blockade" by Eltahir et al., is, in my opinion, a very well written and described original work focusing on a PEA-based approach to identify and quantify several immune-related plasma proteins able to predict the benefit of immunotherapy in NSCLC patients. The methods and statistical analysis have been deeply and well discussed. However, before publication I suggest some minor revision:

  • I strongly suggest to carefully read the text for the presence of many typos and English errors;
  • Please correct and clarify the sentence in row 188 - 192. 

Author Response

Reviewer 2: The manuscript titled "Plasma proteomic analysis in non-small cell lung cancer patients treated with PD1/PD-L1 blockade" by Eltahir et al. is, in my opinion, a very well written and described original work focusing on a PEA-based approach to identify and quantify several immune-related plasma proteins able to predict the benefit of immunotherapy in NSCLC patients. The methods and statistical analysis have been deeply and well discussed. However, before publication, I suggest some minor revision.

Response: We appreciate the positive evaluation.

Reviewer 2: I strongly suggest to carefully read the text for the presence of many typos and English errors.

Response: The manuscript has been revised by a professional English editing service.

Reviewer 2: Please correct and clarify the sentence in rows 188–192.

Response: We have corrected the sentence accordingly.